# Association of Production and Selected Dimensional Conformation Traits in Holstein Friesian Cows

**DOI:** 10.3390/ani14182753

**Published:** 2024-09-23

**Authors:** Zsolt Jenő Kőrösi, Gabriella Holló, Szabolcs Bene, László Bognár, Ferenc Szabó

**Affiliations:** 1National Association of Hungarian Holstein Friesian Breeders, Lőportál Street 16, H-1134 Budapest, Hungary; korosi@holstein.hu (Z.J.K.); bognar@holstein.hu (L.B.); 2Institute of Animal Sciences, Kaposvár Campus, Hungarian University of Agriculture and Life Sciences, Guba Sándor Street 40, H-7400 Kaposvár, Hungary; bene.szabolcs.albin@uni-mate.hu; 3Department of Animal Sciences, Albert Kázmér Faculty, Széchenyi István University, Vár Square 2, H-9200 Mosonmagyaróvár, Hungary; szabo.ferenc@sze.hu

**Keywords:** milk yield, fat yield, protein yield, stature, chest width, body depth, rump width

## Abstract

**Simple Summary:**

Dimensional conformation traits of dairy cows, which are in connection with their individual body features and production, are an important functional trait regulating feed efficiency and energy balance. That is why the scientific importance of these traits has increased recently. In this study, the genetic determination, association, and trend of selected dimensional traits, i.e., stature, chest width, body depth, and rump width, in Holstein Friesian cows were evaluated. Based on the results, according to positive but weak correlations and trends, the examined production and conformation traits did not result in meaningful change in the latter along with the increase in production yield.

**Abstract:**

The objective of this study was to estimate the heritability of dairy production traits and that for dimensional traits and to calculate the correlation between the two heritability values in a Holstein Friesian cow herd bred in Hungary. Data of 15,032 Holstein Friesian cows born in the period 2008–2018 from 666 sires were collected for the study in 6 large dairy herds. Among the conformation traits, stature (ST), chest width (CW), body depth (BD), and rump width (RW), and for production traits, in the first lactation of cows, the 305-day milk yield (MY), milk butterfat yield (FY), and milk protein yield (MY) were evaluated. Heritability estimates of ST, CW, BD, and RW were 0.49, 0.25, 0.31, and 0.30, and those of MY, FY, and PY were 0.40, 0.35, and 0.30, respectively. BD and RW had no phenotypic (b = −0.01) or genetic (b = 0.00–0.01) change. The production traits (MY, FY, PY) increased to a greater extent (b = 2.2–43.3) than the examined conformation traits over time. Consequently, it is indicated that the selection for dairy production did not result in an increase in the studied dimensional traits.

## 1. Introduction

Dimensional conformation traits reflecting the type, frame, skeleton, or main structure of cattle that are related to their body weight are important in Holstein Friesian cattle. The Holstein Friesian is one of the most widespread dairy cattle breeds worldwide. The popularity of this breed is due to its high milk production yields. As a side effect of genetic selection to improve production performance, undesirable traits were also augmented [1].

Conformation traits are very important in breeding programs of dairy cows [2]. The functional conformation traits that influence or facilitate the longevity and reproduction status of dairy cows are the appearance of udder conformation, feet, and legs [3]. From another aspect, there are pure size or dimensional conformation traits, as individual body features such as stature, chest width, body depth, and rump width are closely related to the type of cows [4,5]. The question today, at the present level of milk production, is that the response of genetic selection for milk yields may cause changes in the specific structural features of Holstein Friesian cows [4]. Previously, positive correlations were observed for production and body depth, dairy character, and stature [6].

Dimensional traits, which are associated with several frame or type traits and body weight, regulate feed efficiency and energy balance traits in dairy cattle [7,8]. A heavier, larger sized, larger framed cow needs more nutrient for maintenance, resulting in a worse feed conversion in the same milk yield’s production level. As feed costs constitute a large proportion of the total costs, it is important to consider the frame during selection [9]. Vallimont et al. [10] have reported a strong negative genetic correlation between frame and feed efficiency. In recent decades, the scientific importance of dimensional conformation traits has increased due to its connection to feed intake and efficiency [11]. Haile-Mariam [12], Gruber et al. [13], and Ledinek et al. [11] have come to the conclusion that further increases in the frame of cows, with regard to its unfavorable effect on nutrient efficiency, should not be recommended.

The dimensional traits of cows are related to their milk production yields. According to Ledinek et al. [11], light cows produced less milk than cows of medium or heavier weights. Sieber et al. [14] have found a slight positive phenotypic correlation between frame and milk production. A similar positive genetic correlation result has been published by Tapki and Güzey [15], Zink et al. [16], and Manafiazar et al. [17]. Based on these findings, it can certainly be assumed that selection for milk production can increase the dimensional traits of cows as a result of genetic connections.

Several papers published indicate medium heritability (h^2^) estimates of different conformation traits [7,13,15,16,18,19,20,21,22,23,24]. The combined change in production and type during dairy selection can also result from the fact that the genetic determinants of the two traits are similar, but the frame is inherited somewhat better than milk production yield [15,17,18,19].

The dimensional traits of cows are related not only to their milk production but to several other conformation traits. Weak or moderate positive phenotypic and genetic correlations exist between some conformation traits, such as stature, chest depth, chest width, and rump width. These associations could result in an increase in the frame due to the increase in certain conformation traits [14]. According to Alphonsus et al. [25], the genetic correlations of the body conformation traits (ST, CW, BD, HW, HG, BL, RW) with the individual body features were positive and ranged from 0.179 to 0.854. That is one of the reasons that, according to Miglior et al. [26], conformation traits have been of great interest in the dairy cattle industry for decades.

Selection changes in milk production and certain conformation traits affect each other. According to Alcantara et al. [27], among the conformation traits, heel depth, body depth, and dairy capacity had the highest effect on production, type, and thereby profit in Canadian Holsteins.

There is an association between production yield and conformation of dairy cows. A number of papers have been published showing moderate to strong positive phenotypic and genetic correlation among milk, fat, and protein yields and stature, heart girth, body depth, and rump width of Holstein cows [7,15,16,19,23,24,28]. Conformation of dairy cows is in relation with economically important longevity traits, as well [1,29,30].

Some research results reflect genetic changes in some conformation traits of Holstein Friesian cows. Theron and Mostert [31], performing genetic analysis, have come to the conclusion that cows are becoming taller and more angular, while udder traits have also improved. Carvalho et al. [32] have reported coefficients of regression for some conformation traits. According to their results, the rump height, rump angle, body length, and rump length had a positive, weak increasing trend over time.

Dimensional traits can be measured directly on dairy farms. However, it is difficult to determine the ideal time for measuring or scoring, since they depend on the age, the condition of the cow, and the state of lactation. For this reason, many scientists and experts consider predicting the frame of cows based on body measurements or linear conformation scoring. Enevoldsen and Kristensen [33] have estimated dimensional traits using the body measurements hip height, hip width, and body condition score. Banos and Coffey [34] updated their prediction model on the phenotypic and genetic level, finally resulting in a combination of stature, chest width, body depth, and angularity. Haile-Mariam et al. [12] predicted the morphology of cows from linear conformation traits. Gruber et al. [13] have found that heart girth, belly girth, hip width, and body depth were the best predictors of morphology and body weight. Martins et al. [35] have used three-dimensional cameras for estimating dimensional traits. Cappai et al. [36] have applied integrating individual electronic identification (EID) with 3D digital images for the evaluation of the morphology in Charolais bulls. Ruchay et al. [37] have used a non-rigid 3D shape reconstruction utilizing data from depth cameras. Their results indicate that this approach can serve as a new, accurate method for non-contact body measurement of livestock.

However, in addition to dairy production, several functional and conformation traits are also included as selection criteria in the breeding program of the Holstein Friesian breed. The reduction in frame and some other linear traits can also be a breeding goal due to the improvement of efficiency.

In the Holstein breeding program in Hungary [38], as in other countries, in addition to milk production and some functional traits such as somatic cell count, longevity, and calving ease, among conformation traits, the udder and leg structure are selection criteria. Other conformation traits, such as stature, chest width, body depth, and rump width are not included in the selection index.

An exploration of the recent literature highlights that this field was less investigated in recent years. However, this kind of information would be useful for breeders, scientists, and other specialists. For this reason, according to our opinion, in order to measure efficiency, the analysis of the relationship between production and some conformation traits and their trends might provide new, up-to-date information, helping to modernize breeding programs.

In view of the actual knowledge about the genetic selection for the improvement of production performance, the question as to mass, size, and conformational traits of heifers to improve profitability is open. We hypothesized that the analysis of combined data from genetic, phenotypic, and productive data would contribute to the understanding of modern breeding outcomes.

In our study, we do not intend to deal with the functional conformation, but rather, focus on the most important dimensional conformation traits only. According to the literature cited above, they are in the strongest association with the individual features of cows. As a consequence of this, the main objective of this investigation was to estimate the heritability of dairy production traits and of dimensional traits and to calculate the correlation between the two heritability values in a Holstein Friesian cow herd bred in Hungary in the present. Moreover, we wanted to see clearly whether the current selection program for milk yield production causes a change in stature, chest width, body depth, and rump width, in the important dimensional traits of Holstein Friesian cows.

## 2. Materials and Methods

### 2.1. The Database Used

Data of 15,032 Holstein Friesian cows born in the period 2008–2018 from 666 sires were collected for the study in 6 large dairy farms in Hungary. The cow herds are supervised by the National Association of Hungarian Holstein Friesian Breeders in Hungary. The source of the information for conformation and production traits was available in the database of the above-mentioned association. Table 1 contains the structure of the starting database of the studied Holstein Friesian population.

### 2.2. The Investigated Traits

Among the conformation traits, stature (ST), chest width (CW), body depth (BD), and rump width (RW), and for production traits in the first lactation of cows, 305-day milk yield (MY), milk butterfat yield (FY), and milk protein yield (PY), were processed and evaluated.

The production traits were measured permanently, 10 times in a lactation and adjusted to a 305-day period. Each conformation trait was scored by the same specialist of the association using a 1–9 scale of a linear scoring system.

### 2.3. Estimation of Population Genetic Parameters and Breeding Values

For the estimation, the BLUP animal model was used. According to the study, two matrices were created. One of these was the pedigree matrix and the other was the database matrix. The pedigree matrix of relatives included pedigree data for full sibs, half sibs, sires, dams, and grandparents. The database matrix included all data from effects and traits. In the model, fixed effects were the herd, birth year of the cow, birth season of the cow, and the age (conformation traits) or the age at first calving (production traits). Random effects were the individual (cow).

Cows were categorized by age at scoring to four groups (24.0–27.0, 27.1–30.0, 30.1–33.0, and 33.1–36.0 months). By the age at first calving, four groups (20.0–23.0, 23.1–25.0, 25.1–27.0, and 27.1–34.0 months) were also created. Table 2 shows the applied models for the estimations.

The used basic model was as follows:y = X_b_ + Z_a_ + e(1)
where “y” is the vector of observations; “b” is the vector of fixed effects; “a” is the vector of random animal effects; “e” is the vector of random residual effects; “X” and “Z” are the incidence matrices relating records to fixed and animal effects, respectively.

Breeding values (BVs) for all conformation and production traits of the entire population were estimated. Due to size reasons, these BV results are not presented in this paper. The reliability of the estimated breeding values for the studied traits was 0.70–073 (70–73%). These results encouraged us to use BV for further evaluation.

For the estimation, BLUP animal model MTDFREML [39] software was used.

### 2.4. Phenotypic and Genetic Correlations

Phenotypic correlation (rp) values were calculated between the dimensional conformation and production traits, while the genetic correlation (rg) values were calculated between the BVs of animals in conformation and production traits. For the calculation, SPSS 27.0 (2020) software was used.

### 2.5. Phenotypic and Genetic Trends

During the calculation of the phenotypic trends for the investigated traits, data of cows born in the same year were averaged, and then the mean values were plotted against the year of birth. For fitting function to the resulting set of points, linear regression analysis was used. The dependent variable (Y) was the mean of the traits, and the independent variable (X) was the birth year of the cow. The values of the constant (a), the slope (b), and the fit (R^2^) and their statistical reliability were also determined. The genetic trends of the conformation and production traits—likewise Ostler et al. [40]—were determined from the average BV of animals born in the same year. Two kinds of trends were determined, one from the BV of sires and the other from the BV of the entire population born in the same year.

For the calculation of the genetic trends of the examined traits, a linear regression method was used. BV of sires as well as BV of entire population were evaluated. The annual mean values were the dependent value and the appropriate year was the independent value in the regression method used. Similarly to the phenotypic trend calculation, the values of the constant (a), the slope (b), and the fit (R^2^), as well as their statistical reliability, were determined in this part of the study. The genetic trends were estimated for the period from 1997 to 2015 for sires and from 1996 to 2018 for the entire population.

## 3. Results

Descriptive statistics of the examined traits of the Holstein Friesian cows are summarized in Table 3.

The first lactation production of cows in the studied ten-year period was quite advantageous and balanced, CV% = 18% or below, and favorable. The average MY, FY, and PY were 10,179.4 kg, 380.3 kg, and 333.1 kg, respectively. The conformation data are somewhat more heterogeneous than those for production; the CV% shows a value of 16% to 21%. Table 4 summarizes the different effects on the studied conformation and production traits.

As seen in the table, all examined factors such as sire, herd, year and season of birth, and age of cow, had a significant (*p* < 0.01) effect on both production and conformation traits.

Heritability estimates of the production traits, the MY, FY, and PY, in this study were 0.34, 0.35, and 0.30, respectively. The same values for the conformation traits ranged from 0.25 to 0.49; the lowest value was found for the CW and the highest for ST (Table 5).

Thus, the genetic determination of ST is a little bit stronger, that of CW is weaker, and those of BD and RW seem to be similar to that of the production traits.

Table 6 summarizes the correlation coefficients among conformation and production traits. Obviously, as it was known before, production yield traits are in a strong or very strong positive phenotypic (rp = 0.76–0.94) and genetic (rg = 0.61–0.90) association with each other.

Among the evaluated conformation traits, both the phenotypic (rp = 0.24–0.60) and the genetic (rg = 0.26–0.0.62) correlations are from weakly to strongly positive in each case.

For the association of conformation and production traits, each examined relationship shows a positive but weak or very weak phenotypic (rp = 0.02–0.15) and genetic (rg = 0.01–0.21), and in some cases a non-significant, connection.

From this point of view, there was no difference among the phenotypic and genetic; moreover, in the genetic correlation coefficients depended on which BLUP model they were obtained from.

The phenotypic and genetic trend of the studied conformation and production traits are summarized in Table 7.

The phenotypic change in MY, despite the meaningful increase (b = 43.3), is not significant; however, the genetic trend shows a significantly increasing tendency (b = 5.5 and 16.5) with a weak determination (R^2^ = 0.24 and 0.29). FY shows a little bit higher significant phenotypic increase (b = 2.2) than that of the genetic change (b = 0.3 and 0.5). The phenotypic increase in PY is not statistically proved; however, the genetic increase is similar (b = 0.2 and 0.5) to the increase in FY.

As for the trends of the conformation traits, the CW shows no significant phenotypic and genetic change; however, the trends of ST are significant, but the change is very low over time. The fitting values of the linear function for ST are quite reasonable (R^2^ = 0.44–0.69), but the slope indicates very little change (b = −0.06 to +0.03). Similarly, BD and RW have no phenotypic (b = −0.01) or genetic (b = 0.00–0.01) change over time in the studied ten-year period.

## 4. Discussion

During its genetic progress, the Holstein Friesian cattle breed has achieved high milk production. The big challenge for breeders is how to further raise their production so that the dimensional traits of the cows, which are in relation to the dimensional conformation and feed efficiency traits, do not get worse.

However, according to Battegin et al. [41], there are differences between countries in terms of the scored conformation traits, and there is agreement that dimensional traits are important.

Since the dimensional traits and frame size of cows are related to their dairy production and several other conformation traits, we were curious about the genetic determination, association, and trend of the latter in high-producing herds.

To know more about these relationships, in processing the data of more than 15,000 cows in 6 large Holstein Friesian cow herds in Hungary over a 10-year period, the heritability relationship and trend of the production yield and some dimensional conformation traits were evaluated.

### 4.1. Heritability Estimates

The heritability estimates (h^2^) of the production traits, i.e., MY, FY, and PY, in this study were 0.34, 0.35, and 0.34, respectively, and appeared to be a little bit higher than those reported by Samoré et al. [19] for the same traits for previous years: 0.22, 0.19, and 0.18, respectively. However, Xue et al. [42] have published a higher value (0.47) of heritability estimates for milk yield.

The finding in the present study that h^2^ values of ST, CW, BD, and RW are medium, 0.49, 0.25, 0.31, and 0.30, respectively, aligns with the data published in several sources. Veerkamp [20] found that values for these traits were similarly medium or better and mostly in the range of 0.4 to 0.5. According to Haas et al. [7], the h^2^ estimates for linear type traits of ST, heart girth, BD, and RW were 0.69, 0.38, 0.39, and 0.47, respectively. Tapki and Güzey [15] found h^2^ estimates for ST, CW, BD, and RW of 0.39, 0.31, 0.38, and 0.26, respectively, and Zink et al. [16] found 0.39, 0.28, 0.17, and 0.22, respectively. Xue et al. [42] have reported values of h^2^ of 0.30, 024, and 0.32 for ST, CW, and BD, respectively.

However, some authors have published lower values for some conformation traits than were found in our study. Our results agree only in part with Ahlborn and Dempfle [18], who have reported h^2^ values of ST and other BW-related traits of 0.25 and 0.29 for Holstein Friesian and 0.16 and 0.23 for Jersey. Berry et al. [21] have published heritability estimates for the type traits of 0.11 to 0.43. According to Samoré et al. [19], the h^2^ estimates of ST, BD, and RW were 0.36, 0.25, and 0.15, respectively. Zavadilová and Štípková [22] published that h^2^ estimates ranged from 0.05 to 0.43 for rump traits. Cassandro et al. [23] found that heritability estimates of ST, CW, BD, and RW were 0.199, 0.123, 0.169, and 0.098, respectively. Roveglia et al. [24] reported low heritability for most conformation traits, with the exception of a moderate value (0.32) for stature, in Italian Jerseys.

Based on both our own results and some literature data [7,15,16,19], it seems that the genetic determinations of the evaluated production and conformation traits are similar to each other.

However, corresponding our finding to the aforementioned literature sources, the genetic determination of the ST among the studied conformation traits, due to its higher heritability estimates, seems to be somewhat stronger than that of the CW, BD, and RW.

### 4.2. Genetic Correlations

The findings in this study that production traits are in a strong or very strong positive genetic (rg = 0.61–0.90) association with each other is consistent with several of the literature sources [2,8,14]. No differences were found between the phenotypic and genetic correlation of the counterpart traits. However, Roveglia et al. [24] found that genetic correlations were generally stronger than their phenotypic counterparts in the Italian Jersey.

The results for the association of the ST, CW, BD, and RW with the production traits indicate positive but weak or very weak genetic (rg = 0.01–0.21) relationships. This result agrees with the findings of Samoré et al. [19], who have also published weak positive genetic correlation coefficients of MY, FY, and PY: 0.09, 0.10, and 010 with ST; 0.22, 0.06, and 0.05 with BD; and 0.08, 0.03, and 0.07 with RW, respectively. Kruszyński et al. [28] have found a similar very weak positive genetic correlation of MY with height, depth, and width of rump, such as 0.04, 0.01, and 0.03, respectively. Zink et al. [16] have published genetic correlations of ST with MY, FY, and PY of 0.19, 0.30, and 0.27, respectively; furthermore, they found similar values of the CW, 0.02, 0.05, and 0.04, respectively, and of the BD, 0.19, 0.10, and 0.11, respectively. These results agree with the findings of Khmelnychyl et al. [43], who have found correlation coefficients of milk yield and height (0.211–0.341), body depth (0.282–0.369), and rump width (0.211–0.368). Similarly, Xu et al. [40] have reported rg of 0.34, 0.09, and 0.32 for ST, CW, and BD, respectively.

The findings of the present study are partly different from the result of Ahlborn and Dempfle [18], who have found a little bit higher correlation coefficient; that is, the ST was a moderate, positive genetic correlation with MY, FY, and PY. Similarly, higher positive genetic correlation values were obtained by Berry et al. [21] between type traits and MY.

By contrast, Haas et al. [7] have found that genetic correlations of milk production with conformation traits of lactating Holstein Friesian cows were higher than the aforementioned values, i.e., 0.47, 0.39, 0.48, and 0.26 for ST, heart girth, BD, and RW, respectively. Similarly, Tapki and Güzey [15] have published a stronger genetic correlation of ST and BD with production traits than in our findings.

Comparing our results with data of the literature, we can assess that we have not found as strong an association as was published by several authors. Thus, it could be expected that selection for production might result in less change in the studied conformation traits than in the MY, FY, and PY in the described circumstances.

Agreeing with the results published in some of the literature sources [13,15,21], there were moderate or strong genetic (rg = 0.26–0.62) correlations among the studied ST, CW, BD, and RW traits in our study. These results encourage us to assume that selection for a dimensional conformation trait could alter the other specific structural features as well.

### 4.3. Genetic Trends

Based on the results of the genetic trend evaluation of production yield traits, the slope (b) is positive in each case, indicating increasing production. However, the rise of the PY is quite slight (b = 0.2–0.5). The increase in MY (b = 5.5 and 16.5) and FY (b = 2.2) surpasses the increase in PY. These results, regarding the MY and PY, are consistent with the findings of Haas et al. [7] and Kruszyński et al. [28] but are a little bit different for FY. The latter authors found regression of 10.38, 0.51, and 0.23 for MY, FY, and PY, respectively.

The results of the regression estimation for conformation traits, according to which the slope values of ST, BD, RW, and CW are very low (b = −0.06 to +0.03), indicate only small changes in the dimensional traits of cows over time. Similar results to our study have been obtained by Kruszyński et al. [28], who found regression (b) of height 0.106, trunk depth 0.035, and RW 0.001. Carvalho et al. [33] have reported similarly low values of coefficients of regression for some conformation traits of dairy Gir cows, as follows: rump height 0.04, rump angle 0.01, body length 0.01, and rump length 0.01.

The findings in this study also reflect that the production traits, such as MY, FY, and PY, changed to a greater extent than the studied dimensional conformation traits ST, BD, RW, and CW in the examined ten-year period. Consequently, it indicates that the selection changes for production yields did not result in a significant change in the specific structural features of Holstein Friesian cows in the studied ten-year period.

## 5. Conclusions

Our aim in this study was to see more clearly the heritability of dairy production and dimensional conformation traits and the correlation between them in Holstein Friesian cows in the present. The results indicate that the genetic determination of the milk yield, stature, chest width, body depth, and rump width traits are similar. This finding led to the hypothesis that production and dimensional conformation traits can change together during the selection for milk yield.

However, the positive weak correlations and trends pointed out here did not indicate a change in the dimensional conformation traits of Holstein Friesian cows in Hungary along with the increasing dairy production in the past ten-year period.

## Figures and Tables

**Table 1 animals-14-02753-t001:** The structure of the starting database of the studied Holstein Friesian population.

Starting Parameters	Used Database
Time period of examination, the birth year of cows	2008–2018
Number of herds	6
Number of cows	15,032
Number of the examined sires (sire of cow)	666
Birth date of sires	1997–2015
The average number of female progeny (cow) per sire	22.57
Number of the examined dams (dam of cow)	11,787
Birth date of dams	1996–2017

**Table 2 animals-14-02753-t002:** The applied BLUP animal models for the estimations.

Traits	Conformation Traits	Production Traits
Pedigree matrix		
- animal (cow)	+	+
- sire (sire of cow)	+	+
- dam (dam of cow)	+	+
- full sibs, half sibs	+	+
- grandparents	+	+
Fixed effects		
- herd	+	+
- birth year of cow	+	+
- birth season of cow	+	+
- age of cow at scoring	+	-
- age of cow at first calving	-	+
Examined traits		
- ST	+	-
- CW	+	-
- BD	+	-
- RW	+	-
- MY	-	+
- FY	-	+
- PY	-	+

+ = the model includes this effect; - = the model does not include this effect; ST = stature; CW = chest width; BD = body depth; RW = rump width; MY = 305-day milk yield in first lactation; FY = 305-day milk butterfat yield in first lactation; PY = 305-day milk protein yield in first lactation.

**Table 3 animals-14-02753-t003:** Descriptive statistics of the conformation and production traits of Holstein Friesian cows (N = 15,032).

Trait	$\bar{X}$	SE	SD	CV%	Median	Min	Max	*p* ^#^
Age of cow at conformation scoring, AGE (month)	29.2	0.0	2.6	8.9	28.9	24.0	36.0	0.06
Age of cow at first calving, AFC (month)	24.8	0.0	2.0	8.1	24.5	20.0	34.0	0.07
Lactation interval, LAC (day)	388.0	0.5	62.3	16.1	324.0	200.0	500.0	0.09
Stature, ST (score)	6.1	0.0	1.3	21.5	6.0	1.0	9.0	0.15
Chest width, CW (score)	5.5	0.0	1.0	18.1	6.0	1.0	9.0	0.22
Body depth, BD (score)	5.8	0.0	0.9	15.7	6.0	1.0	9.0	0.26
Rump width, RW (score)	5.3	0.0	1.1	20.7	5.0	1.0	9.0	0.20
305-day milk yield in first lactation, MY (kg)	10,179.4	15.1	1856.6	18.2	10,216.0	5000.0	18,000.0	0.01
305-day milk butterfat yield in first lactation, FY (kg)	380.3	0.6	68.0	17.9	379.7	145.8	648.5	0.01
305-day milk protein yield in first lactation PY (kg)	333.1	0.5	56.4	16.9	334.1	148.5	568.8	0.01

^#^ Kolmogorov–Smirnov test (if *p* > 0.05, the normal distribution is confirmed).

**Table 4 animals-14-02753-t004:** Different effects on the conformation and production traits of the studied Holstein Friesian cows.

Trait	Factors					
	Sire of Cow	Herd	Birth Year of Cow	Birth Season of Cow	Age at Scoring	Age at First Calving
Classes	666	6	11	4	4	4
ST	*p* < 0.01	*p* < 0.01	*p* < 0.01	*p* < 0.01	*p* < 0.01	-
CW	*p* < 0.01	*p* < 0.01	*p* < 0.01	*p* < 0.05	*p* < 0.01	-
BD	*p* < 0.01	*p* < 0.01	*p* < 0.01	*p* < 0.01	*p* < 0.01	-
RW	*p* < 0.01	*p* < 0.01	*p* < 0.01	*p* < 0.01	*p* < 0.01	-
MY	*p* < 0.01	*p* < 0.01	*p* < 0.01	*p* < 0.01	-	*p* < 0.01
FY	*p* < 0.01	*p* < 0.01	*p* < 0.01	*p* < 0.01	-	*p* < 0.01
PY	*p* < 0.01	*p* < 0.01	*p* < 0.01	*p* < 0.01	-	*p* < 0.01

ST = stature; CW = chest width; BD = body depth; RW = rump width; MY = 305-day milk yield in first lactation; FY = 305-day milk butterfat yield in first lactation; PY = 305-day milk protein yield in first lactation.

**Table 5 animals-14-02753-t005:** Population genetic parameters of the conformation and production traits of Holstein Friesian cows.

Traits	Parameters			
	σ^2^_d_	σ^2^_e_	σ^2^_p_	h^2^ ± SE
Stature (ST)	0.78	0.81	1.59	0.49 ± 0.02
Chest (CW)	0.22	0.65	0.87	0.25 ± 0.02
Body depth BD	0.24	0.54	0.78	0.31 ± 0.02
Rump width; RW	0.34	0.78	1.12	0.30 ± 0.02
305-day milk yield in first lactation (MY)	819,426.0	15,906,540.4	2,410,076.4	0.34 ± 0.01
305-day milk fat yield in first lactation (FY)	1163.3	2138.0	3001.3	0.35 ± 0.02
305-day milk protein yield in first lactation (PY)	655.4	1544.8	2200.2	0.30 ± 0.02

σ^2^_d_ = additive direct genetic variance; σ^2^_e_ = residual variance; σ^2^_p_ = phenotypic variance; h^2^ = heritability.

**Table 6 animals-14-02753-t006:** Phenotypic and genetic correlations among the conformation and production traits of Holstein Friesian cows.

r	CW	BD	RW	MY	FY	PY
Phenotypic (r_p_)						
- ST	+0.40 *	+0.43 *	+0.32 *	+0.00	+0.04 *	0.00
- CW		+0.60 *	+0.24 *	+0.02 ^&^	+0.09 *	+0.05 *
- BD			+0.26 *	+0.10 *	+0.15 *	+0.10 *
- RW				+0.13 *	+0.08 *	+0.12 *
- MY					+0.76 *	+0.94 *
- FY						+0.81 *
Genetic (based on BV of sires) (r_g_)						
- ST	+0.30 *	+0.43 *	+0.41 *	+0.07	+0.11 *	+0.04
- CW		+0.62 *	+0.26 *	+0.01	+0.09 ^&^	+0.09 ^&^
- BD			+0.35 *	+0.08 ^&^	+0.17 *	+0.11 *
- RW				+0.04	+0.04	+0.04
- MY					+0.62 *	+0.89 *
- FY						+0.68 *
Genetic (based on BV of entire population) (r_g_)						
- ST	+0.34 *	+0.43 *	+0.40 *	+0.10 *	+0.11 *	+0.10 *
- CW		+0.62 *	+0.27 *	+0.01	+0.09 *	+0.08 *
- BD			+0.32 *	+0.13 *	+0.21 *	+0.15 *
- RW				+0.10	+0.10 *	+0.10 *
- MY					+0.61 *	+0.90 *
- FY						+0.69 *

* *p* < 0.01; ^&^ *p* < 0.05; BV = breeding value; ST = stature; CW = chest width; BD = body depth; RW = rump width; MY = 305-day milk yield in first lactation; FY = 305-day milk butterfat yield in first lactation; PY = 305-day milk protein yield in first lactation.

**Table 7 animals-14-02753-t007:** Phenotypic and genetic trends in the conformation and production traits of Holstein Friesian cows.

Trend	Y	Slope			Intercept			Fitting	
		b	SE	*p*	a	SE	*p*	R^2^	*p*
ST									
- P	aP	−0.06	0.01	<0.01	120.60	25.65	<0.01	0.69	<0.01
- GSB	aBV	+0.03	0.01	<0.01	−52.80	14.53	<0.01	0.44	<0.01
- GAB	aBV	+0.01	0.00	<0.01	−21.15	3.75	<0.01	0.60	<0.01
CW									
- P	aP	−0.01	0.01	NS	23.80	11.28	<0.10	0.25	NS
- GSB	aBV	+0.00	0.00	NS	−8.29	6.25	NS	0.10	NS
- GAB	aBV	+0.00	0.00	NS	−2.33	1.57	NS	0.11	NS
BD									
- P	aP	−0.01	0.01	<0.10	30.56	13.4	<0.05	0.27	<0.10
- GSB	aBV	+0.01	0.00	NS	−11.33	7.56	NS	0.12	NS
- GAB	aBV	+0.00	0.00	<0.01	−4.75	1.62	<0.01	0.30	<0.01
RW									
- P	aP	−0.06	0.01	<0.01	120.99	20.61	<0.01	0.78	<0.01
- GSB	aBV	+0.01	0.01	<0.05	−24.89	11.33	<0.05	0.23	<0.05
- GAB	aBV	+0.01	0.00	<0.01	−16.22	3.20	<0.01	0.56	<0.01
MY									
- P	aP	+42.3	24.8	NS	−74,870.7	49,850.5	NS	0.25	NS
- GSB	aBV	+16.5	6.2	<0.05	−32,974.8	12,517.8	<0.05	0.29	<0.05
- GAB	aBV	+5.5	2.3	<0.05	−10,968.5	4522.3	<0.05	0.24	<0.05
FY									
- P	aP	+2.2	0.6	<0.01	−3993.3	1161.8	<0.01	0.61	<0.01
- GSB	aBV	+0.5	0.3	<0.10	−989.9	526.1	<0.10	0.18	<0.10
- GAB	aBV	+0.3	0.1	<0.01	−530.2	168.7	<0.01	0.32	<0.01
PY									
- P	aP	+1.6	0.8	<0.10	−2980.9	1696.0	NS	0.30	<0.10
- GSB	aBV	+0.5	0.2	<0.05	−986.6	401.5	<0.05	0.27	<0.05
- GAB	aBV	+0.2	0.1	<0.01	−400.8	114.2	<0.01	0.39	<0.01

NS = not significant; ST = stature; CW = chest width; BD = body depth; RW = rump width; MY = 305-day milk yield in first lactation; FY = 305-day milk butterfat yield in first lactation; PY = 305-day milk protein yield in first lactation; *p* = phenotypic trend; GSB = genetic trend based on BV of sires; GAB = genetic trend based on BV of entire population; aP = average phenotypic data of the trait; aBV = average breeding value.

## Data Availability

The datasets analyzed or generated during the study are available in the database of the National Association of Hungarian Holstein Friesian Breeders at https://www.holstein.hu/research_data.html (accessed on 18 September 2024).

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
