# Peer review of "Association of Production and Selected Dimensional Conformation Traits in Holstein Friesian Cows"

_animals, 2024, doi:10.3390/ani14182753_

Round 1
Reviewer 1 Report
Comments and Suggestions for Authors
-
The study uses a large and diverse dataset, encompassing 15,032 Holstein-Friesian cows over a decade (2008-2018). This extensive dataset enhances the reliability and generalizability of the findings.
-
By estimating heritability and phenotypic and genetic correlations, the paper addresses important genetic aspects of cattle breeding. Understanding these correlations is crucial for making informed breeding decisions.
-
The study covers both production traits (milk yield, butterfat yield, protein yield) and body weight-related conformation traits (stature, chest width, body depth, rump width), offering a holistic view of the factors influencing dairy cow performance.
-
The paper highlights the trends in production and conformation traits over time, which is valuable for understanding the impact of selection practices in modern dairy farming. The finding that production traits have increased more than body weight-related conformation traits could have significant implications for breeding strategies and feed efficiency.
-
The study’s methodology, including the use of heritability estimates and trend analysis, is robust and appropriate for the research objectives. This methodological approach provides a solid foundation for the conclusions drawn.
-
While the study focuses on body weight-related conformation traits, it does not consider other important conformation traits such as udder characteristics or leg structure, which are also critical for dairy cow performance and longevity.
-
The paper mentions feed efficiency indirectly, but it does not include direct measurements or analyses related to feed efficiency. This omission limits the ability to draw definitive conclusions about the relationship between production traits and feed efficiency.
-
The study does not account for environmental factors that might influence the traits being studied. Factors such as farm management practices, climate, and nutrition could have a significant impact on both production and conformation traits.
-
The data is derived from 666 sires, which could introduce bias if certain sires were overrepresented or if selection practices were not uniform across the herds. This potential bias could affect the generalizability of the heritability estimates and correlations.
-
The study is limited to Hungarian Holstein herds, which might restrict the applicability of the findings to other regions with different environmental and management conditions. This geographic limitation reduces the global relevance of the study.
-
While the paper identifies trends in production and conformation traits, it does not delve deeply into the potential reasons behind these trends. A more detailed discussion of the underlying factors driving these changes would strengthen the analysis.
Overall, the paper offers valuable insights into the relationship between production and conformation traits in Holstein-Friesian cows, but it could benefit from a broader scope and a more in-depth exploration of certain aspects.
Author Response
Dear Reviewer,
Thank you very much for your appreciation of the manuscripts in notes 1-5., and the useful critical comments in notes 6-11.
Our reply are as follows:
Note 6: While the study focuses on body weight-related conformation traits, it does not consider other important conformation traits such as udder characteristics or leg structure, which are also critical for dairy cow performance and longevity.
Reply: Agree with the reviewer that some other conformation traits (udder characteristics or leg structure etc.) are very important and crucial for longevity. These traits were not objective of this study, because we just wanted to evaluate the effect on frame size, body weight related, morphometric traits. We have a plan to prepare another manuscript based on the association of longevity and different conformation traits.
Note 7: The paper mentions feed efficiency indirectly, but it does not include direct measurements or analyses related to feed efficiency. This omission limits the ability to draw definitive conclusions about the relationship between production traits and feed efficiency.
Reply: Unfortunately the feed intake is not recorded trait, consequently no direct measure for feed efficiency. We were able to see he cahange of frame, body weight of cows. Our hypothesis was that larger framed, heavier cows have higher maintenance nutrient requirement, align with worst feed conversion.
Note 8: The study does not account for environmental factors that might influence the traits being studied. Factors such as farm management practices, climate, and nutrition could have a significant impact on both production and conformation traits.
Reply: Agree with the reviewer that the study did not accounted all environmental factors, for example the climate effect.
However, the BLUP model used for breeding value estimation (estimation of genetic effects) included farm, herd, birth year of cow, birth season of cow , age and, or the age at first calving as fixed effects.. Some other possible effects was included in the residual effect. The reliabilities of the results were quite reasonable (70-73%), therefore the results were significant.
Note 9: The data is derived from 666 sires, which could introduce bias if certain sires were overrepresented or if selection practices were not uniform across the herds. This potential bias could affect the generalizability of the heritability estimates and correlations.
Reply: Agree with the reviewer that sire effect could have been influenced bias if certain sires were overrepresented. For luck the BLUP animal model, applied for breeding value estimation in this study, can handle the different sire effects minimizing the bias as sire effect was included in the model. The statistical analyses showed significant results both for population genetic parameters and production and conformation traits.
Note 10: The study is limited to Hungarian Holstein herds, which might restrict the applicability of the findings to other regions with different environmental and management conditions. This geographic limitation reduces the global relevance of the study.
Reply: Agree. However, we hope that some other researchers in different environments will do similar study. The results from different surroundings would allow us to make a global conclusion.
Note 11: While the paper identifies trends in production and conformation traits, it does not delve deeply into the potential reasons behind these trends. A more detailed discussion of the underlying factors driving these changes would strengthen the analysis.
Reply: Agree and thank you. We tried to make the discussion part of this study more detailed in the improved manuscript based on your and other reviewer’s suggestions.
Reviewer 2 Report
Comments and Suggestions for Authors
Quite an interesting topic. Unfortunately, the body weight of cows was not determined, and this feature appears many times.
Only conformation characteristics were assessed subjectively and body weight was discussed on this basis. This is inappropriate.
The structural features being assessed should be focused on and discussed.
Author Response
Dear Reviewer,
Thank you very much for your comments and useful suggestions.
Our reply is as follows:
Note: Unfortunately, the body weight of cows was not determined, and this feature appears many times. Only conformation characteristics were assessed subjectively and body weight was discussed on this basis. This is inappropriate.
Reply: Agree that body weight of cows was not determined, or registered in the database. For this reason, we tried to determine the change in body weight indirectly, from the change of certain conformation traits. Sometimes body weight was mentioned, but more times body weight related conformation traits were written in the manuscript.
Following your and the other reviewer's advices the term body weight (BW) has been deleted from manuscript and morphometric conformation traits are used instead of it.
Note: The structural features being assessed should be focused on and discussed.
Reply: Thank you for your advice. The manuscript has been improved focusing by discussion and conclusion on structural features.
Reviewer 3 Report
Comments and Suggestions for Authors
Dear Authors,
many thanks for this valuable piece of work. I read it with great interest and I found merit in it. Congratulations for the strenght points and practical results from your experimental activity. I find your paper a contribution of reference in this field (which is also my field of interest).
The title is fine but I would suggest to use the term "selected" in place of "some". It sounds more scientific, if authors agree.
The simple summary start very well, but I feel that the long sentence LL. 17-19 should be rephrased. Maybe, splitting into shorter and less informative (meaning with general concepts) sentences would be a benefit for the simple summary. You have the chance to explain further in the abstract, in detail.
ABSTRACT: May I suggest to rephrase as: "The objective.... estimate the correlation between heritability of genetic traits and expression in the phenotype accounting production yields and selected body weight-related morphometric measures...". Instead of "our days", please prefer "present or actual lines".
L. 25: Again, I would recommend to refer to "among selected linear measures, stature (height at rump*), ..."
*I know that in practice the rump height is used in dairy cows, but did you also considered height at withers?
With body depth (BD), do you mean chest depth?
It is my wish that author would provide a figure of an ideal cow where all morphometric measures used in this experiments were used. It would be very informative to summarize the heritability coefficient, and the positive (though mild, but significant) correlation coefficient with milk production and composition. The core-message of your paper would be readable at a glance.
L. 33: I believe that feed efficiency should be better explained.
L. 38: ".... is one of widest spread dairy cattle breeds ..." and delete in dairy production, you will end the sentence with worldwide.
L. 39: please, add after production "yields".
L. 39: "As a side effect of genetic selection to improve production performance, also undesirable traits were augmented ...".
L. 50: Please, prefer as follows ".... should not be recommended".
L. 59: Can you report the value(s) as to heritability coefficient of BW?
L. 65: please, correct the citation form and rephrase the sentence
L. 76: this is repeated. Same concept expressed earlier.
L. 79: mind the brackets, turn the round one into square bracket.
L. 80: Please, shift citation '1', before 22,23.
LL. 82-84: Please, make the sentence clearer than this, I would suggest to shorten the sentences or split into two, instead of using very long ones. Sometimes, the reader misses the meaning.
L. 97: Please add also the papers by Cappai, M. G., Gambella, F., Piccirilli, D., Rubiu, N. G., Dimauro, C., Pazzona, A. L., & Pinna, W. (2019). Integrating the RFID identification system for Charolaise breeding bulls with 3D imaging for virtual archive creation. PeerJ Computer Science, 5, e179. and Ruchay, A., Kober, V., Dorofeev, K., Kolpakov, V., & Miroshnikov, S. (2020). Accurate body measurement of live cattle using three depth cameras and non-rigid 3-D shape recovery. Computers and Electronics in Agriculture, 179, 105821.
L. 107: Please, prefer to start: The exploration of recent literature highlights that this field was less investigated in recent years.
L. 109: Not only to breeders! (just a comment!).
L. 114: May I dare to suggest to improve your paragraph? I would say (sorry for being so invasive...): "In view of the actual knowledge about the genetic selection for the improvement of production performance, the question as to mass, size and conformational traits of heifers to improve profitability is open. We hypothesized that the analysis of combined data from genetic, phenotypic and productive data would contribute to the understanding of modern breeding outcomes. The objective of this investigation was... (and repeat the ones in the amended abstract).
M&M are fine.
Results are fine and clear.
Discussion:
L. 269: Please, prefere to write "...than those reported by Samoré et al.... and delete the part (published...)"
L. 308: Please, prefer the expression "... agree only in part with Ahlborn..."
L. 312: Please, prefer "By contrast, Haas...."
L. 317: Please, correct the grammar and prefer "....Comparing our results with data of the literature, we can assess...."
L. 322-325: Please, rephrase the whole sentence with grammar and style correction.
L. 327: Please, reconsider the use of the term "Whereas". Is that what you really mean?
L. 336: "ChangeS"
L. 341: "Reflect_" Please, be sharp in correcting grammar error. You may let your paper to proofreading.
CONCLUSION
LL. 347-350: This is hard to understand. The sentence should be totally rephrased.
LL. 351-354: PLease, see the above comment. Rephrase, it is hard to follow.
Conclusions should be totally rewritten, and avoid to recall the lack of paper. You should give here a recommendation and that is enough.
Comments on the Quality of English Language
Dear Editors
the paper is written by several "hands". It is clear that some writers are skilled, but some are not and some tracts of the text are difficult to understand.
I suggested for extensive editing of english language.
Author Response
Dear Reviewer,
Many thanks for your appreciation and useful suggestions.
Note: The title is fine but I would suggest to use the term "selected" in place of "some". It sounds more scientific, if authors agree.
Reply: The word "some" was replaced by "selected". Moreover, since academic editor and reviewer 1 suggested the term “body weight related conformation traits” to replace by “morphometric conformation traits” in the title and in the other parts of the manuscript. So, the new title: Association of production and selected morphometric conformation traits in Holstein-Friesian cows.
Note: The simple summary start very well, but I feel that the long sentence LL. 17-19 should be rephrased. Maybe, splitting into shorter and less informative (meaning with general concepts) sentences would be a benefit for the simple summary. You have the chance to explain further in the abstract, in detail.
Reply: The long sentence LL. 17-19 has been rephrased and shortened.
Note: Abstract: May I suggest to rephrase as: "The objective.... estimate the correlation between heritability of genetic traits and expression in the phenotype accounting production yields and selected body weight-related morphometric measures...". Instead of "our days", please prefer "present or actual lines"
Reply: The sentence has been replaced as follows:
The objective of this study was to estimate the correlation between. heritability of genetic traits and expression in the phenotype accounting production yields and selected morphometric measures of Hungarian Holstein cow herds in actual lines.
The term: body weight related conformation traits, has been deleted and replaced by morphometric measures according to the suggestion of academic editor and reviewer 1.
Note: L. 25: Again, I would recommend to refer to "among selected linear measures, stature (height at rump*), ..."
*I know that in practice the rump height is used in dairy cows, but did you also considered height at withers?
Reply: The part of sentence has been changed to „among selected linear measure traits”
Agree with the reviewer that height at withers is as useful trait as height at rump. Unfortunately it is not registered in Hungary, similarly to the practice of some other countries. It does not included in the guideline of World Holstein Friesian Federation.
Note: With body depth (BD), do you mean chest depth?
Reply: No. Body depth (BD) and chest depth (CD) are different traits. BD is measured or scored at back, while CD at withers. CD is not recorded, it does not including in the guideline of World Holstein Friesian Federation.
Note: It is my wish that author would provide a figure of an ideal cow where all morphometric measures used in this experiments were used. It would be very informative to summarize the heritability coefficient, and the positive (though mild, but significant) correlation coefficient with milk production and composition. The core-message of your paper would be readable at a glance.
Reply: Agre with the reviewer that a figure of an ideal cow would be very informative, and the core-message of this paper would be readable at a glance. However, we have evaluated only four morphometric traits, which are not enough for this purpose. We think that there should be a multitude of conformation traits (angularity, rib-,udder-, leg structure etc.) to figure of an ideal Holstein cow.
Note: L. 33: I believe that feed efficiency should be better explained.
Reply: A sentence: „A heavier, larger sized, larger framed cow need more nutrient for maintenance, resulting a worse feed conversion in the same milk production yield level” has been inserted.
Note: L. 38: ".... is one of widest spread dairy cattle breeds ..." and delete in dairy production, you will end the sentence with worldwide.
Reply: The sentence has been modified as follows: The Holstein-Friesian is one of widest spread dairy cattle breeds worldwide.
Note: L. 39: please, add after production "yields"
Reply: Word „yields” is inserted.
Note: L. 39: "As a side effect of genetic selection to improve production performance, also undesirable traits were augmented ..."
Reply: The part of the sentence has been replaced to: As a side effect of genetic selection to improve production performance, also undesirable traits were augmented.
Note: L. 50: Please, prefer as follows ".... should not be recommended".
Reply: „Should not be” has been inserted instead of cannot be
Note: L. 59: Can you report the value(s) as to heritability coefficient of BW?
Reply: h2 = 0.45-0.65 has been inserted.
Note: L. 65: please, correct the citation form and rephrase the sentence
Reply: The citation has been coreccted and the sentence rephrased as follows: These associations could result in increased the BW by rising of certain conformation traits (Sieber et al.) [9]
Note: L. 76: this is repeated. Same concept expressed earlier.
Reply: Repeating sentence has been deleted.
Note: L. 79: mind the brackets, turn the round one into square bracket.
Reply: Brackets has been changed.
Note: L. 80: Please, shift citation '1', before 22,23.
Reply: It has been shifted in the improved manuscript.
Note: LL. 82-84: Please, make the sentence clearer than this, I would suggest to shorten the sentences or split into two, instead of using very long ones. Sometimes, the reader misses the meaning.
Reply: The sentence has been shortened and probably made clearer.
Note: L. 97: Please add also the papers by
Cappai, M. G., Gambella, F., Piccirilli, D., Rubiu, N. G., Dimauro, C., Pazzona, A. L., & Pinna, W. (2019). Integrating the RFID identification system for Charolaise breeding bulls with 3D imaging for virtual archive creation. PeerJ Computer Science, 5, e179. and
Ruchay, A., Kober, V., Dorofeev, K., Kolpakov, V., & Miroshnikov, S. (2020). Accurate body measurement of live cattle using three depth cameras and non-rigid 3-D shape recovery. Computers and Electronics in Agriculture, 179, 105821.
Reply: The mentioned two papers has been added and cited.
Note: L. 107: Please, prefer to start: The exploration of recent literature highlights that this field was less investigated in recent years.
Reply: The starting sentence has been replaced by the suggested one.
Note: L. 109: Not only to breeders! (just a comment!).
Reply: Words scientists and other specialists has been inserted.
Note: L. 114: May I dare to suggest to improve your paragraph? I would say (sorry for being so invasive...): "In view of the actual knowledge about the genetic selection for the improvement of production performance, the question as to mass, size and conformational traits of heifers to improve profitability is open. We hypothesized that the analysis of combined data from genetic, phenotypic and productive data would contribute to the understanding of modern breeding outcomes. The objective of this investigation was... (and repeat the ones in the amended abstract).
Reply: The original text has been replaced by the sentences suggested.
Note: L. 269: Please, prefer to write "...than those reported by Samoré et al.... and delete the part (published...)"
Reply: The sentence has been modified according to the suggestion.
Note: L. 308: Please, prefer the expression "... agree only in part with Ahlborn..."
Reply: The sentence has been improved according to the suggestion.
Note: L. 312: Please, prefer "By contrast, Haas...."
Reply: The sentence has been started with "By contrast, Haas....
Note: L. 317: Please, correct the grammar and prefer "....Comparing our results with data of the literature, we can assess...."
Reply: Correction has been made as follows: "....Comparing our results with data of the literature, we can assess...
Note: L. 322-325: Please, rephrase the whole sentence with grammar and style correction.
Reply: The wole sentence has been rephrased and improved.
Note: L. 327: Please, reconsider the use of the term "Whereas". Is that what you really mean?
Reply: The word whereas has been deleted.
Note: L. 336: "ChangeS"
Reply: Letter „s” has been put to the end of word.
Note: L. 341: "Reflect_" Please, be sharp in correcting grammar error. You may let your paper to proofreading.
Reply: Grammar errors have been corrected, and the paragraph is improved by skilled in English.
Note: LL. 347-350: This is hard to understand. The sentence should be totally rephrased.
- 351-354: Please, see the above comment. Rephrase, it is hard to follow.
Conclusions should be totally rewritten, and avoid to recall the lack of paper. You should give here a recommendation and that is enough.
Reply: The sentence mentioned has been improved. Conclusions have been totally rewritten.
Round 2
Reviewer 2 Report
Comments and Suggestions for Authors
It's better, but you still have to think to get something specific.
Four features of cow structure (stature, chest width, body depth, rump width) were taken into account. Why were these chosen and not others? In the review, do not write about body weight, but about individual body features and their importance in cattle breeding.
When discussing the obtained results, please focus on specific structural features and less on the size of the cows and their weight.
No significant conclusion was reached.
Author Response
Respond to the reviewer 2 (Second version of manuscript)
Manuscript ID: animals-3212346
Type of manuscript: Article
Title: Association of production and selected dimensional conformation traits in Holstein-Friesian cows
Authors: Zsolt JenÅ‘ KÅ‘rösi, Gabriella Holló *, Szabolcs Bene, László Bognár, Ferenc Szabó
Dear Reviewer,
Thank you again for your comments, helpful and useful suggestions regarding our modified manuscript.
Our reply to your comments and suggestions are as follows.
Note: It's better, but you still have to think to get something specific.
Reply: The next text has been inserted to the appropriate place of the manuscript:
Conformation traits are very important in dairy cows. There are functional conformation traits that influence or facilitate the longevity and reproduction status of dairy cows are the appearance of udder conformation, feet and leg. From another aspect there are pure size or, dimensional conformation traits, such as stature, chest width, body depth, rump width, as special body features that are closely related to the type of cows. The latter traits are important since they regulate feed efficiency and energy balance.
The question today, in the present level of milk production, is that the response of genetic selection for milk yields may cause change in the specific structural features of Holstein cows.
Note: Four features of cow structure (stature, chest width, body depth, rump width) were taken into account. Why were these chosen and not others?
Reply: Manuscript, at the end of the introduction, is completed by next:
In our study, we do not intend to deal with the functional conformation, but only focus on the most important pure size conformation traits to see the structural features. According to the literature, cited in the manuscript, they are in the strongest association with the type cows.
Note: In the review, do not write about body weight, but about individual body features and their importance in cattle breeding.
Reply: Agree, thank you. The term „body weight” has been deleted from the manuscript. Instead of it individual body features, or dimensional traits are used.
The dimensional conformation traits, such as stature, chest width, body depth, rump width, are important because they are closely related to as special body features of cows. The latter traits are useful since they regulate feed efficiency and energy balance.
Note: When discussing the obtained results, please focus on specific structural features and less on the size of the cows and their weight.
Reply: Thank you. Instead of size and body weight of cows we have focused on the specific structural features of cows in the improved manuscript, in the discussion chapter.
Note: No significant conclusion was reached.
Reply: The conclusion chapter of the manuscript has been rewritten as follows:
The positive weak correlations and trends, pointed out here, have not indicated change of the dimensional, morphometric conformation traits together with increasing milk production in the past ten year period.
However, the findings in this study, that the genetic determination of the production traits and the dimensional conformation traits are similar, call attention to the possible change of the specific structural features along with the selection for milk yield in the future.
Dear Reviewer,
We would like to ask for your acceptance of our response and the revised manuscript.
Reviewer 3 Report
Comments and Suggestions for Authors
Dear Authors,
I thank you for welcoming all my requests of change. I realized that you amended the manuscript and I have no other suggestion. From my side, it is ready to be considered for publication.
Author Response
Dear Reviewer,
Thank you very much for your review report. Many thanks for your acceptance.